# AGs-Unet: Building Extraction Model for High Resolution Remote Sensing Images Based on Attention Gates U Network

**DOI:** 10.3390/s22082932

**Published:** 2022-04-11

**Authors:** Mingyang Yu, Xiaoxian Chen, Wenzhuo Zhang, Yaohui Liu

**Affiliations:** 1School of Surveying and Geo-Informatics, Shandong Jianzhu University, Jinan 250101, China; ymy@sdjzu.edu.cn (M.Y.); 2020160106@stu.sdjzu.edu.cn (W.Z.); 2Hebei Key Laboratory of Earthquake Dynamics, Sanhe 065201, China

**Keywords:** AGs-Unet model, deep learning, high resolution remote sensing image, building extraction, WHU dataset

## Abstract

Building contour extraction from high-resolution remote sensing images is a basic task for the reasonable planning of regional construction. Recently, building segmentation methods based on the U-Net network have become popular as they largely improve the segmentation accuracy by applying ‘skip connection’ to combine high-level and low-level feature information more effectively. Meanwhile, researchers have demonstrated that introducing an attention mechanism into U-Net can enhance local feature expression and improve the performance of building extraction in remote sensing images. In this paper, we intend to explore the effectiveness of the primeval attention gate module and propose the novel Attention Gate Module (AG) based on adjusting the position of ‘Resampler’ in an attention gate to Sigmoid function for a building extraction task, and a novel Attention Gates U network (AGs-Unet) is further proposed based on AG, which can automatically learn different forms of building structures in high-resolution remote sensing images and realize efficient extraction of building contour. AGs-Unet integrates attention gates with a single U-Net network, in which a series of attention gate modules are added into the ‘skip connection’ for suppressing the irrelevant and noisy feature responses in the input image to highlight the dominant features of the buildings in the image. AGs-Unet improves the feature selection of the attention map to enhance the ability of feature learning, as well as paying attention to the feature information of small-scale buildings. We conducted the experiments on the WHU building dataset and the INRIA Aerial Image Labeling dataset, in which the proposed AGs-Unet model is compared with several classic models (such as FCN8s, SegNet, U-Net, and DANet) and two state-of-the-art models (such as PISANet, and ARC-Net). The extraction accuracy of each model is evaluated by using three evaluation indexes, namely, overall accuracy, precision, and intersection over union. Experimental results show that the proposed AGs-Unet model can improve the quality of building extraction from high-resolution remote sensing images effectively in terms of prediction performance and result accuracy.

## 1. Introduction

Rapidly developing remote sensing technology provides massive data for urban planning, mapping, and disaster management. As real estate resources in urban and rural areas, buildings are of great significance in both urban dynamic monitoring and suburban construction inspection [1,2,3,4] and a precise and immediate extraction of buildings becomes critical [5,6].

With the continuous improvement in the spatial resolution of optical remote sensing images, building detection from high-resolution images has attracted increasing attention. The data volume of early remote sensing images is small, and the extraction of buildings in images mainly relies on artificial recognition, visual interpretation and vector feature extraction [7,8]. In recent years, as the remote sensing image data gradually show the characteristics of being massive, multi-scale and multi-sourced [8,9,10], the cost of building contour extraction is high and the quality cannot be measured by a unified standard with manually interpreting the image. Therefore, various methods for building extraction from remote sensing images are proposed by related researchers in feature extraction based on prior knowledge such as shadow, edge, roof color, shape design, etc. [11,12]. Some methods, including template matching [13], mathematical morphology [14,15], active contours [16,17], graph theory [18,19], random forests [15] and support vector machines [20,21], are based on building roof detection. Prior knowledge, including the complexity and diversity of building shape, roof surface, imaging conditions and spatial environment, has been used to easily confine the building shape areas. Although previous methods for building extraction from high-resolution remote sensing images have made some progress, these methods based on artificial design features generally have shortcomings, such as low extraction accuracy, complex processing, and insufficient feature utilization. These methods also need various rules to manually predefine features, thereby leaving the extraction of buildings from large-scale remote sensing images still difficult.

At present, the methods for extracting buildings from high-resolution remote sensing images mainly uses deep learning developed in computer vison. Compared to the above mentioned traditional methods of building detection and semantic segmentation, deep learning methods have advantages in automatically extracting high-dimensional features of image maps. The convolution neural network (CNN) [22] has developed rapidly and is widely used in fields such as natural image classification, target detection, and semantic segmentation. Compared with the traditional method of extracting target features based on artificial design, the CNN model can automatically extract and fully explore the characteristics of the input image and has strong feature representation ability. At present, common CNN models mainly include AlexNet [23], VGGNet [24], GoogLeNet [25], and ResNet [26], however, these CNN models cannot directly produce accurate building contours. Therefore, it is necessary for segmented maps to preprocess the image in certain circumstances, and the full convolution networks (FCNs) provide a deep learning framework for end-to-end image semantic segmentation by transforming the full connection layer of the CNN into the convolution layer, which greatly improve the training and prediction efficiency of the model.

At present, FCNs have been used in building extraction from aviation and satellite images [27,28,29]. For example, Ji et al. [30] proposed a Siamese U-Net model with two branches and shared weights. The model input includes the original image and its down-sampling feature map. Through the training of multi-source data sets, including aerial images, satellite images, raster, and vector labels, the extraction effect of large buildings improved significantly. To extract buildings with high precision, Guo et al. [31] proposed a multi-loss neural network based on attention. The model can use a single attention module to improve the sensitivity of building objects in remote sensing images and reduce the influence of irrelevant background regions. The multi-loss method composed of base and offset losses can reduce the test loss value and further enhance the effect of semantic segmentation of the model. To improve the robustness of the building extraction model of remote sensing images, Zhou et al. [28] constructed the pyramid self-attention network model of the “end-to-end” neural network. The local features of the building are obtained by the backbone network. The color, texture, and advanced semantic features of the building were extracted by the pyramid self-attention module, which improved the accuracy and integrity of the single building extraction. Although the deep learning model has achieved the end-to-end extraction of building contours, the accuracy and stability of this task need to be improved further so that it can be effectively applied to actual production and life.

In the segmentation module, two main problems need to be solved at the present stage. On one hand, the high-dimensional features have less sensitivity to the response of background information and ground truth, which results in the loss of spatial information in the target region. On the other hand, boundary ambiguity is aroused by the convolutional segmentation method of considering the intersection ratio (IoU) only. For semantic segmentation, top-down architectures such as the FCN8s [32], SegNet [33], U-Net [34], and DANet models [35] have been proposed to further improve the performance and efficiency. Generally, the development of these models is driven by the spatial information input to the high-dimensional processor after several aggregations of the convolutional layers, which will lead to poor edge precision of segmentation results and a blurring of the boundary. This network structure does not perform well in the semantic segmentation of buildings, mainly due to the loss of the spatial information of local and regions as well as the poor, stride-convolved, and high-dimensional features. Therefore, some methods are proposed to compensate for local information [36,37] and improve the selection of the number of feature channels. Among them, PISANet model [28] obtains the global features and comprehensive features of the buildings through the pyramid self-attention module, which makes full use of the spatial information in the image; ARC-Net model [27] reduces the computational cost and shrinks the model size by residual blocks with asymmetric convolution, which improves the extraction effect based on reducing the model parameters.

In order to solve the above problems, a lightweight spatial attention module, Attention Gate (AG), has been proposed [38] and applied to building extraction task by fine-tuning the structure of the AG module. In this work, we further consider the applicable efficiency of the AG module in the building extraction task of remote sensing images, and appropriately adjust the position of ‘resample’ in the AG module. Above all, we aim to explore the effectiveness of a recent AG for building features extraction in remote sensing images, and a novel Attention Gates U network (AGs-Unet) model is further presented. AGs-Unet integrates four attention gates with a primeval and single U-Net architecture, in which a series of AG units are integrated to the ‘skip connection’ for highlighting salient feature information. AGs-Unet makes use of four AGs, which cannot only catch the large-scale features, but also pay more attention, to a certain extent, to small buildings as well. The integration of AG modules shows effectiveness in different and complex environments on the task of building extraction. In this experiment, we also compare AGs-Unet with the traditional FCN8s [32], SegNet [33], U-Net [34], DANet model [35], the state-of-the-art PISANet [28], and ARC-Net [27] models to carry out the experimental comparison of remote sensing image building automatic extraction task on both the WHU Dataset and the INRIA Aerial Image Labeling Dataset in terms of prediction accuracy, parameter number, and training time.

## 2. Materials and Methods

High-resolution remote sensing images have the characteristics of numerous ground objects, complex ground background, and complex data information. Many different targets have certain similarities, which blur the boundary of the target area and produce “noise” by fully studying the basic deep learning model in the field of high-resolution remote sensing image processing. This study proposes a grid-based AGs module, which consists of four AG modules. AGs were originally used in machine translation [39,40], image interpretation [41], and other fields to improve the effect of text processing by providing different weights to various local texts. AGs that act on image analysis are a channel selection mechanism [31,42] that simulates the relationship between channels through computational efficiency to enhance the feature extraction ability of the whole network to extract target features and construct an attention map more effectively [43]. Inspired by convolutional block attention modules (CBAM) [44] and squeeze-and-excitation networks (SENet) [45], this paper develops the AGs for dense buildings in high-resolution remote sensing images and explores the impacts of AG on areas with different percentages of buildings. Meanwhile, we construct a grid-based AGs module to enhance the sensitivity of the grid-based AGs to small- and medium-sized buildings in the feature map.

### 2.1. AGs-Unet Model Framework

The AGs-Unet model, which is the attention control model constructed by adding AGs modules to a standard U-Net architecture, is shown in Figure 1. The deep learning model based on U-Net architecture has good computational performance and can efficiently utilize GPU memory, which is widely used in the field of image segmentation. The attention gate control mechanism is beneficial for highlighting the effective features of the target and suppressing redundant invalid information in the image multi-scale feature extraction task. Therefore, the AGs-Unet model constructed in this paper sets the advantages of the U-shaped structure and attention control mechanism. The model proposed in this paper can capture the wide-area information of the context at the low-dimensional scale and extract the global coarse-scale features of the image. It can also extract the abstract fine-level features of the image in high-dimensional scale and highlight the location and boundary of the building in the feature map through the attention valve. Among them is the gridding, multi-scale extraction of feature maps through the ‘jump connection’ access to the decoder part, with both coarse and fine levels of dense building prediction fusion.

Figure 1 shows that AGs-Unet is a U-shaped symmetric deep learning model that directly outputs the building extraction results from one end of the image input to the other end. It fully shows that the deep learning method has the advantages of speed, convenience, economics, and end-to-end without intermediate processing. The specific structure of AGs-Unet can be divided into three parts:(1)The encoder, which is composed of four convolution blocks, can extract features at different levels using global and local context information (*x_i_* represents the feature of layer *i*). Each convolution block consists of two convolutional (Conv) layers, two batch normalization (BN) layers, and two rectified linear unit (ReLU) activation function layers. The Maxpooling layer in the model can extract the maximum value of the local area in the feature map and construct it into a new feature map, which helps reduce the number of parameters in the model and prevent overfitting;(2)The converter is composed of the fifth convolution block and AGs in four ‘skip connection’ processes. The fifth convolution block includes Conv, Maxpool, and BN, which abstracts the feature map from the encoder to the highest level. The number of channels of the feature map is superimposed to 1248, the size is reduced to 16 × 16, and the abstract feature map of the highest dimension is extracted. The AGs stringing in the ‘jump connection’ filter out the feature points in the low-dimensional feature map that are beneficial in extracting the building and in filtering and suppressing irrelevant features and nodes. Four AG extract effective features from all aspects and dimensions in four different grid dimension levels from low to high. The converter connects the feature map that corresponds to the encoder and decoder and solves the problem of the disappearance of the reverse propagation gradient to some extent;(3)The decoder; the number of channels of the corresponding feature map is adjusted by convolution operation, and the size of the fused feature map is expanded by up-sampling and gradually fused with the multi-level feature map and restored to the size of the original input map, since the underlying high-dimensional feature map began to integrate into the feature map of the corresponding size in the encoder.

Visibly, the encoder contains the same number of convolution blocks as the decoder. AGs-Unet samples the features introduced by the converter step-by-step through up-sampling and obtains the feature map with the same size as the input image. Finally, the channel number of the model is adjusted by 1 × 1 convolution, and the final building segmentation results in the remote sensing image obtained by a Sigmoid activation function. The specific parameters of AGs-Unet are shown in Table 1.

### 2.2. Grid-Based AGs Module under U-Net Architecture

#### 2.2.1. Attention Mechanism

The attention mechanism from human intuition, which aims to select more critical information for current tasks, has been widely used in various sequential learning tasks. The core steps of attention mechanism include the following: first, the importance scores of each candidate vector are calculated, and then the scores are normalized to weights. Finally, these weights are applied to the candidate vector to generate attention results. Many attention mechanism models are applied to deep convolutional neural networks to optimize the feature extraction process. For example, the CBAM [44] infers the attention weight from the input feature map along the two dimensions of space and channel. Subsequently, this weight was multiplied with the input feature map to realize the adaptive adjustment of features. DANet [35] extracts the attention weights of both sides of the input feature map by parallel processing the location attention module that can enhance the global feature fusion and the channel attention module that are conducive to strengthening the correlation of local semantic features. SENet [45] shows the correlation between learning channel characteristics through global average pooling operation. The global context network [46] uses a self-attention mechanism to model the relationship between query pairs and integrates non-local and SENet modules to complete global context modeling.

Therefore, a lot of work has been performed to extend self-attention to computer vision for image target recognition and semantic segmentation, which inspired us to apply an attention mechanism to remote sensing image processing to extract buildings from high-resolution remote sensing images.

#### 2.2.2. AG Module

The attention coefficient α ∈ [0, 1] is designed in AG to identify prominent image regions and suppress irrelevant feature responses to retain and activate neurons that are only related to buildings to the greatest extent. The multiplication of the input feature map and α in the encoder at the pixel level is the output of AG: ${\hat{x}}_{i}^{l}{=x}_{i}^{l}\;\cdot {\;\alpha }_{i}^{l}$. Usually, the calculation of the attention value of a single scalar needs to use the value of each pixel vector $x_{i}^{l}$, and where $x_{i}^{l}\;\in {\;R}^{F_{l}}$, and $F_{l}$ represents the number of feature maps in the layer l. The activation function ReLU in the model is expressed as: $\sigma 1(x_{i,}^{l}){=\max\;(0,\;x}_{i}^{l})$, where *i* represents the spatial dimension.

This study only extracts buildings from remote sensing images, which are semantic segmentation of a single category. Therefore, this model only designs the attention coefficient. As shown in Figure 2, the valve control gate vector $g_{i}\;\in {\;R}^{F_{g}}$ acts on each pixel *i* to determine the focusing region. The control gate vector contains context information to remove low-level feature responses. This study draws on the additional attention mechanism [31] proposed in machine translation to obtain the control coefficient, and the additional attention mechanisms is expressed by Formulas (1) and (2).
(1)$$q_{\mathrm{aun}}^{l}{(x}_{i}^{l}{,\;g}_{i}{;\;\Theta }_{\mathrm{att}}{)=\psi }^{T}(\sigma_{1}{(W}_{x}^{T}x_{i}^{l}{+W}_{g}^{T}g_{i}{+b}_{g})){+b}_{\psi }\;,$$
(2)$$\alpha_{i}^{l}{=\sigma }_{2}(q_{\mathrm{aun}}^{l}{(x}_{i}^{l}{,\;g}_{i}{;\;\Theta }_{\mathrm{att}}))\;,$$
where $\sigma_{2}{(x}_{i})=\frac{1}{{1+\exp(-x}_{i})}$ represents the Sigmoid activation function. The feature of AG is that a set of parameters $\Theta_{\mathrm{att}}$ contains: linear transformation $W_{x}\;\in {\;R}^{F_{l}{\times F}_{\mathrm{int}}}$, $W_{g}\;\in {\;R}^{F_{g}{\times F}_{\mathrm{int}}}$, $\psi \;\in {\;R}^{F_{\mathrm{int}}\times 1}$ and bias parameter $b_{\psi }\;\in {\;R}^{F_{\mathrm{int}}}$, $b_{g}\;\in {\;R}^{F_{\mathrm{int}}}$, and $F_{\mathrm{int}}$ represents the input layer. The linear transformation changes the number of channels of the input tensor through 1 × 1 × 1 channel convolution.

The AG mechanism is detailed as follows, the weight is multiplied by the feature in the attention layer, and the weight is multiplied by the control vector in the valve control layer. ReLU is used to activate the result of the addition of the two multiplication results and the bias. Then, the number of channels is adjusted by linear transformation, and sigmoid is used for activation. The attention coefficient is adjusted in the Resampler. Finally, the original input attention layer feature is multiplied by the activated result, and the result with the same size of the attention layer feature is outputted. In the task of image classification [47], the Softmax activation function is used to normalize the attention coefficient. The disadvantage is that multiple uses of Softmax will result in the activation sparsity at the output end, which results in the disappearance of the gradient. Therefore, this paper only uses one sigmoid activation function in the AGs of the model, so that the AGs parameters have better training convergence. In the proposed grid attention method, the control signal is not a global single vector of all image pixels, but a grid signal suitable for image spatial information. As shown in Figure 1, the control signals of each jump connection aggregate information from multiple dimensions, which increase the grid resolution of the query signal and achieve better performance.

The realization of AGs model includes three parts, namely, *W_g_* of the training gate control coefficient, *W_x_* of the training attention coefficient, and *P_s_* of connecting the two parts to adjust the number of output channels. The structure blocks and parameters of AGs are shown in Table 2. AGs module is helpful in eliminating the necessity of a building location module in multi-level CNNs, reducing the number of parameters and training time of the model, and improving the operation efficiency of the model.

In this paper, grid-based AGs module is added to the standard U-Net architecture to extract the coarse-scale features of multi-dimensional wide-area obtained by jump connection and eliminate the ambiguity of irrelevant information and noise response. Executing the grid-based AGs in the connection operation can more effectively activate the relevant useful nodes in the neural network. In addition, the grid-based AGs module purposefully filtered neurons without activation in forward and backward propagation. The gradient that started from the background region is continuously weighted downward in the reverse propagation, and the shallow model parameters can be updated continuously based on the spatial region related to the building extraction task.

#### 2.2.3. U-Net Architecture with AGs Module

Due to the complexity of multiple ground objects in high-resolution remote sensing images, reducing the number of false positive and false negative pixels in the image for the small target segmentation task of building extraction with large shape changes is still difficult. To improve accuracy, some of the current segmentation frameworks [48] rely on determining the building location boundary first and dividing the task into separate positioning and subsequent segmentation. This paper proves that the location and segmentation of building targets can be achieved by integrating AGs into a standard CNN model without training multiple models and additional model parameters. Different from the localization model in multi-stage CNN, AGs suppress the feature response in irrelevant background regions in multiple dimensions and do not need to prune the region of interest among networks. Therefore, this study integrates AGs into the U-Net structure to further improve the accuracy of building contour extraction in remote sensing images.

Ronneberger et al. [34] proposed a U-Net architecture with symmetric encoding and decoding that fully combines the CNN and the up-sampling part. The number of up-sampling parts and down-sampling parts are equal, and the model is combined by the jump connection between the convolution layer and the deconvolution layer. The jump connection in the U-Net network combines the good characteristics of contraction and expansion path. In multidimensional AGs, $\alpha_{i}^{l}$ corresponds to a vector at each grid scale. In each AG sub-module, complementary information is extracted and fused to define the output of jump connection. To reduce the number of training parameters and computational complexity in AG, this study performs linear transformation (1 × 1 × 1 convolution) without spatial information support, maps the input features to the resolution of the valve control signal through down-sampling, and then decodes the feature map by the corresponding linear transformation, and maps it to the low-dimensional space to complete the attention control operation.

## 3. Experimental Datasets and Evaluation

### 3.1. Dataset

#### 3.1.1. The WHU Dataset

The WHU building dataset [30] is composed of aerial and satellite datasets, and the aerial image dataset is used in this study. The aerial image data are obtained from the New Zealand Land Information Service website, which covers 450 square kilometers of land on the ground. The ground resolution of the image is 0.3 m, and approximately 22,000 independent buildings in the Christchurch area are selected. The dataset provides shapefile format and rasterized data of buildings. The aerial image dataset includes 8189 high-resolution remote sensing images with 512 × 512 pixels. In this study, the PIL module in Python is used to expand the original data into 11,642 images by random rotation, and the samples are divided into three parts, including a training set (8679 images, around 70%), a validation set (1927 images, around 20%), and a test set (1036 images, around 10%). Figure 3 shows an original image and its corresponding label, where black represents the background, and red represents the building.

#### 3.1.2. The INRIA Aerial Image Labeling Dataset

The second dataset used in this research is the INRIA Aerial Image Labeling Dataset [49]. The INRIA dataset covers the world, including Austin, Chicago, Kitsap, Western Tyrol, Vienna, Bellingham, and San Francisco. The spatial resolution of each image is 0.3 m with a size of 5000 × 5000 pixels and surface coverage of 1500 × 1500 m^2^. Following previous investigations [11,28], we selected the first five images of each city for validation and the rest for training. Only two semantic classes were considered as the ground truth; buildings, and non-buildings. An example of an input image and its corresponding label are presented in Figure 4.

### 3.2. Evaluation Criterion

#### 3.2.1. Evaluation Metrics

In this study, the effectiveness and accuracy of each model are verified by experiments. Overall Accuracy (*OA*) in Formula (3), Precision (*Precision*) in Formula (4) and Intersection over Union (*IoU*) in Formula (5) are defined as the indicators of model performance evaluation.
(3)$$\mathrm{Overall}\;\mathrm{Accuracy}\;(\mathrm{OA})=\frac{\mathrm{TP}+\mathrm{TN}}{\mathrm{TP}+\mathrm{TN}+\mathrm{FP}+\mathrm{FN}}\;,$$
(4)$$\mathrm{Precision}=\frac{\mathrm{TP}}{\mathrm{TP}+\mathrm{FP}}\;,$$
(5)$$\mathrm{IoU}=\frac{\mathrm{TP}}{\mathrm{TP}+\mathrm{FP}+\mathrm{FN}}\;,$$
where *TP* is the true positive case, *FP* is the false positive case, *FN* is the false negative case, and *TN* is the true negative case. *OA*: the ratio of the number of pixels correctly classified to the total number of test pixels; *Precision*: the percentage of the number of pixels correctly classified as positive in all predicted positive pixels; *IoU*: the accuracy of the segmentation level.

#### 3.2.2. Model Complexity

In this study, the model complexity is evaluated by the number of parameters and computation of the statistical model [50,51]. Model parameters refer to the total number of parameters required to define the model, that is, the storage space needed to store the model. The amount of computation required by the model is the amount of occupancy calculation required for a given level of data to pass through the network, that is, the amount of computation required when using the model. The parameter values calculate the sum of the weights and the number of offset weights, and Formula (6) is used to calculate the number of parameters on the convolution layer.
(6)$${(K}_{h}{\;\times \;K}_{w}{\;\times \;C}_{in}{)\;\times \;C}_{out}{\times \;C}_{out}\;,$$
where $K_{h}$ represents the length of the convolution kernel, $K_{w}$ represents the width of the convolution kernel, ${\;C}_{in}$ represents the number of input channels, and ${\;C}_{out}$ represents the number of output channels.

Given that neural networks have quite a lot multiplication and addition operations, the computation of the model is measured by the Multiply Accumulate Operations (MAC) of Formula (7).
(7)$${num}_{p}{\;\times \;size}_{out},$$
where ${num}_{p}$ represents the number of parameters in this layer, and ${\;size}_{out}$ represents the size of the output feature diagram.

The formula of a 3 × 3 convolution kernel on the feature graph is Formula (8), which includes nine multiplications and additions, denoted as 9Mac.
y = w [0] × x [0] + w [1] × x [1] + w [2] × x [2] + … + w [8] × x [8],(8)

## 4. Experiment

### 4.1. Experimental Settings

The experiment is based on Pytorch deep learning framework and uses an open resource python module, such as TorchVision, Skimage, and Matplotlib modules, to process images. The computer hardware is equipped with a display card NVIDIA GeForce GTX 3070 Ti, with 8 GB memory, and CUDA 11.0 is used to accelerate the calculation. Due to the limitation of GPU memory, all images after data enhancement are randomly cut to 256 × 256 pixels for model training and cross validation of each epoch.

In the process of experiment super parameter setting, this study carried out many comparative experiments to select the optimal model parameters. Model training uses the Adam optimizer [52] with an initial learning rate of 0.0001. To avoid overfitting, L2 regularization is introduced into all convolutions, and the weight attenuation [53] is set to 0.0001. To adapt to the computer GPU memory constraints, these models’ training inputted eight images per batch and trained for 200 epochs on the WHU dataset, while these models trained for 150 epochs on the INRIA dataset. Among them, the changes in training accuracy and loss value of AGs-Unet in both datasets are shown in Figure 5a,b. During the training, the model accuracy gradually converges to above 0.95 in the fluctuation. While accuracy increases, the loss value gradually decreases and tends to be stable. If not specified, other comparison models are trained in the same hardware and software environment as above.

### 4.2. Experimental Result

In this study, four groups of test data, each group with 40 images, are selected in terms of the proportion of buildings in the whole image to verify the training results of the optimal model and to validate the difference of the extraction accuracy of buildings with different degrees of aggregation. Two representative images from each group are selected for qualitative evaluation, the quantitative accuracy evaluation is also conducted by averaging the *OA* and *IoU* of all images from each group. The images with different proportions of buildings in the whole image from 0–25% are selected as group 1, from 25–50% as group 2, from 50–75% as group 3, and from 75–100% as group 4.

#### 4.2.1. Results of Qualitative Analysis

In this study, U-Net and AGs-Unet models are used to analyze each group of the test set to validate the effect of the AG module, as shown in Figure 6. In the first group of test data, in which the proportion of buildings is small, (a) several scattered small buildings in U-Net and AGs-Unet cannot be clearly identified, and for several relatively large buildings in (a), the AGs-Unet model gets more accurate results than U-Net in building edges extraction. In (b), for a small number of clustered building joints, AGs-Unet predicts clear contours, while U-Net incorrectly identifies the gaps between buildings as buildings. Comparing (a) and (b), it can be seen that the AGs-Unet model can aggregate more valid information by AGs; they avoid invalid features like building gaps; and have better prediction results than the U-Net model with a smaller proportion of buildings.

In the test data of the groups 2 and 3, with a middle proportion of buildings, the distribution of buildings in group (c) is biased toward the local area at the bottom right of the image, and the local AGs-Unet at the regional boundary is more accurate than the building contour predicted by U-Net. Both models below the group (d) misjudge a relatively regular square as a building, however, AGs-Unet has a clearer prediction of the outline of the square, and it is not difficult to find that the target here also has difficulty in judging a building by visual interpretation. In group (e), the proportion of buildings is larger and scattered around the region. AGs-Unet has fewer misclassifications on the edges of the buildings compared to U-Net in the subtle areas. In group (f), AGs-Unet misjudged the square in the middle of the building as a building and failed to identify the target correctly and clearly. In the comparison between groups 2 and 3, when the proportion of buildings in the region is in the middle, the model proposed in this study can basically extract the building outlines. However, there are still difficulties in identifying the plazas with the 25–75% proportions of multiple buildings.

In group 4 of the test data, the proportion of buildings is above 75%, the smaller dispersed buildings in group 4(a) and larger buildings in group 4(b) both make up the main part of the area. The results show that both U-Net and AGs-Net predict that the interiors of medium and small buildings are accurately filled, however, AGs-Unet can extract the whole contour and internal area more completely for large buildings, while U-Net predicts that the interior of the buildings are in part a region of local false negatives. Comparing the experiments of four groups with different proportions of buildings, this study finds that the model is most ineffective in extracting building contours in the region with a small proportion of buildings, and the accuracy of model prediction would increase with the proportion of buildings. However, both models did not further improve the visualization performance of building prediction when the proportion increased from always to the maximum. In regions with different proportions of buildings, the AGs-Unet model proposed in this study can aggregate the visual information of each dimension through AGs, fully adapt to remote sensing image regions with different proportions of buildings, and achieve better prediction results than the U-Net model.

#### 4.2.2. Results of Quantitative Analysis

On the group data, the comparison results of the test accuracy of U-Net and AGs-Unet models are shown in Table 3. To make the data representative, Accuracy and IoU are the average values of all the test data of each group. The accuracy of the AGs-Unet model increased by 6% in group 1, by 0.2% in group 2, by 1.6% in the group 3, and by 0.4% in group 4. The intuitive changes shown in Figure 7b show that AGs-Unet has the highest accuracy in extracting small buildings from images with the least proportion of buildings in the region, however, AGs-Unet has less accuracy than U-Net in extracting large buildings from images with the largest proportion of buildings in the region. The IoU in the AGs-Unet model increased by 6% in group 1, by 1.3% in group 2, by 8.1% in group 3, and by 0.2% in group 4, and its intuitive changes, as shown in Figure 7a, have increased to some extent in each group.

## 5. Discussion

### 5.1. Model Comparison

#### 5.1.1. Comparison of Prediction Results

In Figure 8, the first two columns show the comparison of building extraction results of SegNet, FCN8s, DANet, U-Net, PISANet, ARC-Net, and AGs-Unet models on the WHU dataset, while the third and fourth columns show the results on the INRIA dataset. Overall, compared to the other six models, AGs-Unet have better smoothness on the edge of the building on the test data, and the buildings extracted by AGs-Unet have more accurate edges and fewer internal voids under the action of grid-based AGs modules. In the first image of the test dataset, the buildings in the upper left corner of the image framed by a purple rectangle are selected. The SegNet model with the highest computational resource occupancy is better at extracting the integrity of buildings than FCN8s and DANet. The U-Net model extracts the problem of empty holes in buildings. PISANet model extracts small buildings with poor results in terms of local deficiencies and lack of integrity. The smoothness and accuracy of the edge line extracted by ARC-Net model in the upper left corner of small buildings are not high. The AGs-Unet model extracts the smooth, accurate edge lines, and completes the internal structure of buildings.

In the second image of the test set, two details of the convex part of the extension and the concave part of the subsidence in the whole building selected by the orange frame are selected. The SegNet model prediction map does not extract the convex local information of the building, and large holes in the concave local area exist. Both FCN8s and DANet fail to extract the convex local information of the building, and the FCN8s model’s performance had less missing for concave local feature area, which showed better prediction effect. PISANet and ARC-Net fail to clearly extract the two details of building contour lines. The U-Net model extracts the convex local information of the building. However, a problem of incomplete target prediction in the concave local area exists. The AGs-Unet model accurately extracts convex and concave details. According to the overall analysis of the validation set data, the integrity of buildings extracted by SegNet and FCN8s models is insufficient, and the edge fluency is not high. The AGs-Unet model extracts relatively smooth and accurate edge lines and relatively complete internal structure of buildings, so it has the best experimental effect.

In the INRIA dataset, DANet, ARC-Net and AGs-Unet models have better integrity and the AGs-Unet model is relatively more accurate for the building outlines. The module of double attention in DANet enhances the effect of image feature extraction and makes the integrity of building extraction results better. Meanwhile, the module of AG in AGs-Unet also improves the model′s ability to extract building features, suppresses irrelevant feature information, and facilitates the integrity of building extraction results.

Table 4 shows the quantitative evaluation results of models in the WHU dataset and INRIA dataset. The proposed AGs-Unet model holds the best scores relative to established models, except for *OA*, where DANet performs better in the INRIA dataset. In the WHU dataset, IoU is emphasized and shows that this model is 8% higher than the SegNet model, 7.9% higher than the FCN8s model, 6.5% higher than the DANet model, 1.2% higher than the U-Net model, and 0.8% higher than PISANet, and 0.5% higher than ARC-Net. In the INRIA dataset, analysis of IoU shows that this model is 26.9% higher than model SegNet, 19.8% higher than model FCN8s, 0.6% higher than model DANet, 1.1% higher than model U-Net, 3.1% higher than model PISANet, and 0.3% higher than model ARC-Net.

#### 5.1.2. Model Complexity Comparison

In this paper, experiments are designed according to the method in Section 3.2.2, and the parameters and computations of each model are counted to evaluate the complexity of the model. To measure the time cost of training models, we recorded the average time required for each model to train an epoch on both datasets. As shown in Table 5, the largest number of parameters is FCN8s 134.27 M, AGs-Unet model parameters of these seven models at a medium level. The calculation amount and training time consumption of FCN8s are also the highest, followed by AGs-Unet model. Although the model accuracy has been significantly improved, the calculation consumption has also increased.

This study counts the number of parameters and the amount of computation consumed by each structure in the AGs-Unet model from the encoder, the converter, to the decoder. Figure 9 shows the number of parameters and the amount of computation accumulated layer-by-layer from the bottom to the top. The fifth structure block in the converter has 14.16 M data volume, which accounts for 40.6% of the total data volume, indicating that the image feature transfer converter is the region with the largest amount of model data. The calculation amount of each model is counted in the order of data dimension 3 × 224 × 224. The calculation amount is mainly consumed in the upper sampling stage. The average consumption of ‘Up’ is 5.5 G Mac per time, which accounts for 10.9%, and a total of four ‘Up’ accounts for more than 40%. In the model calculation statistics, the average consumption of each AGs is 0.21 G Mac, accounting for 0.4%.

### 5.2. Ablation Experiment

In this section, the model simplification experiment is performed to ensure that each group is carried out under the same experimental conditions. Considering the number of AGs in the model, this study constructs three other models with U-Net, AGs-Unet-2, and this model and conducted three groups of comparative experiments. The AGs-Unet-2 model represents the removement of AGs in the first and third layers based on the AGs-Unet model, keeping only the two AG modules on the second and fourth layers, hence the name is AGs-Unet-2. To reduce the consumption of computer GPU resources, these three groups of experiments train 50 periods on the basis of other unchanged hyperparameters. The accuracy of the statistical model and IoU are compared with the prediction results of the optimal model on the test set data.

Figure 10 shows the visualization results of the models in three different scenarios of the test set. In the first line, the scene is a small building target with discrete distribution. The U-Net model has obvious missed buildings, while the results of the AGs-Unet-2 model have some finely distributed false detection buildings. This AGs-Unet model has worse results and more accurate building edges. The second scenario focuses on large building targets. U-Net and AGs-Unet-2 models have many missed detections at the edge of buildings, while the buildings extracted by the AGs-Unet model have smoother edges, indicating that the four-dimensional AG module effectively enhances the ability of building edge extraction. As shown in the third line, water background exists in the building target with regular distribution of aggregation, as shown in the upper left area of the image in the third row. In this case, other models have worse performance, resulting in lots of misjudgment. This model correctly excludes the interference of the water area and accurately extracts the building. It is proved that the multi-dimensional features extracted by AGs-Unet through four-level attention gates can suppress background information and extract effective features of buildings.

The test set of the experimental results uses 120 images of the three groups with different proportions of selected buildings. As shown in Table 6, the experimental results show that the number of AGs will affect the accuracy of the test data. The highest accuracy of the AGs-Unet is 97.81%, and the highest accuracy on IoU is 84.38%. In *Precision*, U-Net model is the best, AGs-Unet-2 is the second best, and our proposed AGs-Unet with four AG is lower. We analyze that adding the AG module to standard U-Net increases the complexity of the model, but instead weakens the ability of the model to recognize the false positive pixels. It makes the pixels of *FP* in our proposed AGs-Unet model increase, which decreases the *precision* in turn, as Formula (4) shown.

In short, in the experiment, this study compared the proposed method with the most advanced method and verified the effectiveness and rationality of the proposed method. Thereafter, the WHU building data set was used for model simplification, and remarkable results are shown in the test and verification sets.

## 6. Conclusions

In this study, the AGs-Unet model is proposed, and four AG modules are integrated into the standard U-Net model to solve the task of extracting dense building targets in the semantic segmentation model of high-resolution remote sensing images. Two public datasets, Inria Aerial Image Labeling Dataset and WHU Building Dataset, were used to verify the effectiveness of AGs-Unet. These datasets contain buildings with highly diverse sizes, types and shapes. Experimental results have demonstrated that the AGs-Unet made full use of the local and global information together and obtained higher final accuracy. AGs-Unet can reduce the number of false positive and false negative pixels in the buildings and extract the building contour more accurately. In addition, this paper discusses the advantages and problems of CNN models with different numbers of AGs in remote sensing image building extraction.

In this study, the above experiments prove the effectiveness and progress of the present method, however, it still has some limitations. For example, buildings in areas with sparse distribution of objects have poor extraction and low accuracy, and the method is adversely affected by impermeable surface interference, low quality images caused by clouds and fog, and vegetation shading. Meanwhile, the boundaries of adjacent buildings are blurred, and the method outlined in this paper finds that the model has more difficulty in determining the adjacent relationship of the buildings with blurred boundaries, and are likely to be identified as a single building. We add the partial structure of four AG modules to the connection layer, which increases the numbers of parameters to a certain extent and can raise extra training time. In the future, we intend to introduce interferometric synthetic aperture radar (InSAR) technology to overcome the influence of vegetation shading and non-building interference, yield higher-resolution remote sensing images and improve the building blocks of the host model. We will continue to integrate additional and higher-resolution remote sensing images, and explore the post processing methods to make the relationships of buildings clearer, then improve the extraction accuracy of sparse buildings.

## Figures and Tables

**Figure 1 sensors-22-02932-f001:**
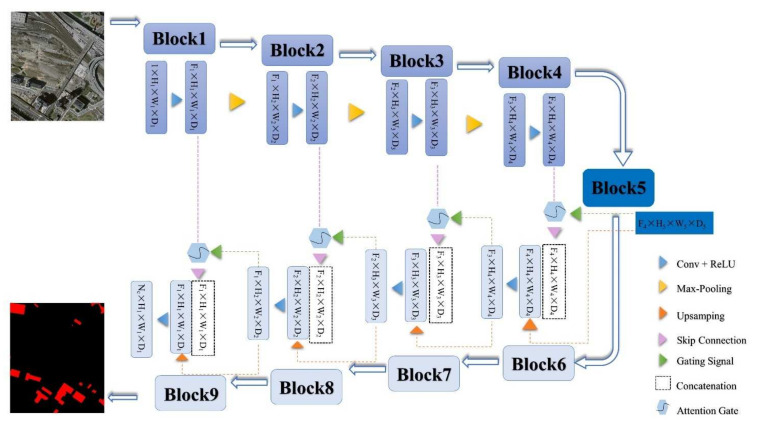
Model framework of AGs-Unet, consisting of three parts: encoder part (blue, block 1–4), the converter (navy blue and pink, block 5 and ‘skip connection’), and decoder (blocks 6–9).

**Figure 2 sensors-22-02932-f002:**
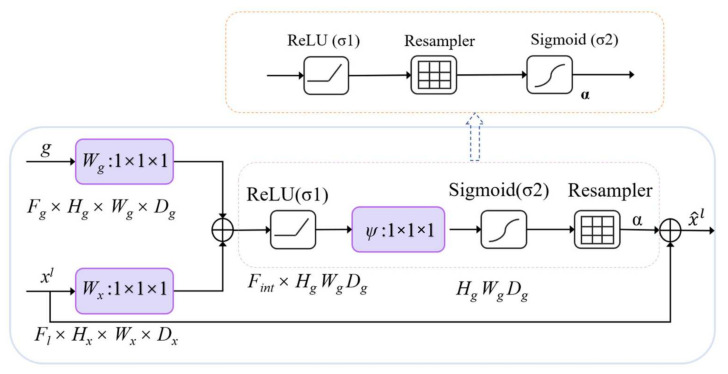
Structure of a AG module. The below shows the original AG module, and above is the part of the AG module changed position of the ‘Resampler’.

**Figure 3 sensors-22-02932-f003:**
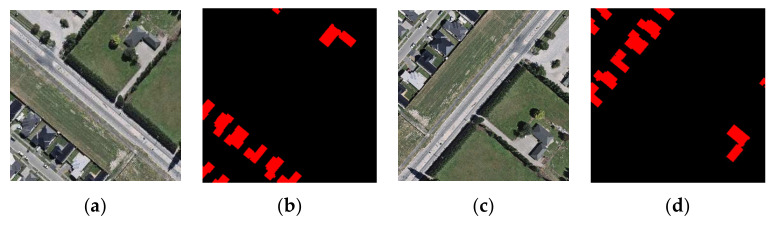
Image and label selected from WHU dataset: (**a**,**b**) show the image and label of the original images in the dataset, respectively; and (**c**,**d**) show the image and label after random rotation, respectively.

**Figure 4 sensors-22-02932-f004:**
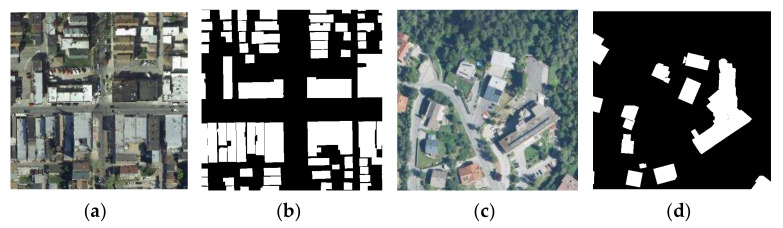
Image and label selected from the INRIA dataset: (**a**,**c**) show the image and (**b**,**d**) show the corresponding label in the dataset, white and black pixels mark building and non-building, respectively.

**Figure 5 sensors-22-02932-f005:**
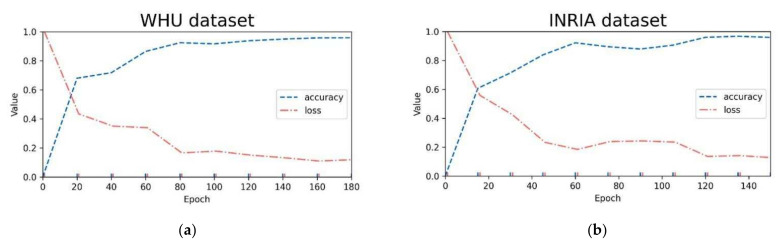
Variation in training accuracy and loss value of AGs-Unet in: (**a**) WHU dataset; and (**b**) INRIA dataset.

**Figure 6 sensors-22-02932-f006:**
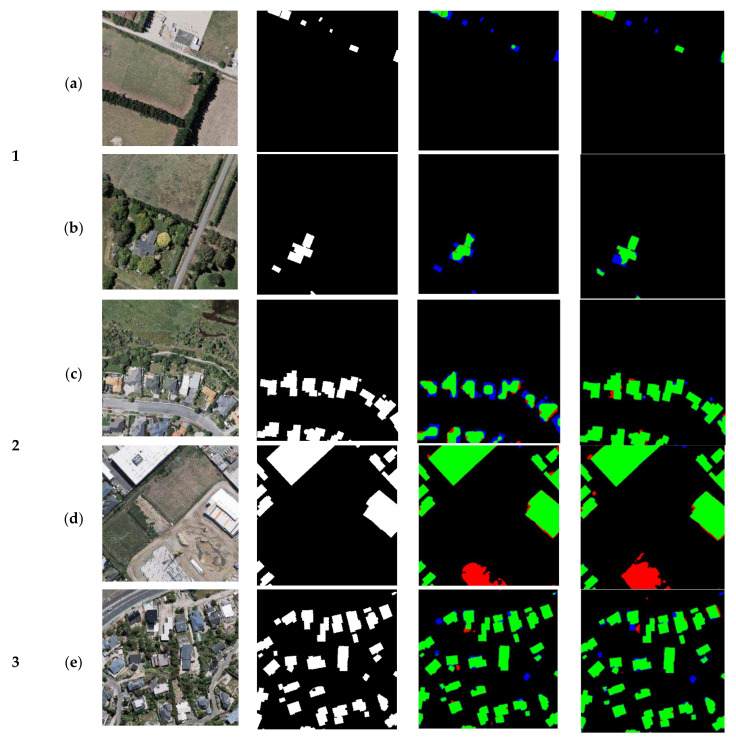
Experimental visualization results of each group. Each team selected two representative images to test the model trained by U-Net and AGs-Unet, where green represented buildings correctly extracted, blue represented buildings missing, red represented buildings incorrectly extracted, and black represented background.

**Figure 7 sensors-22-02932-f007:**
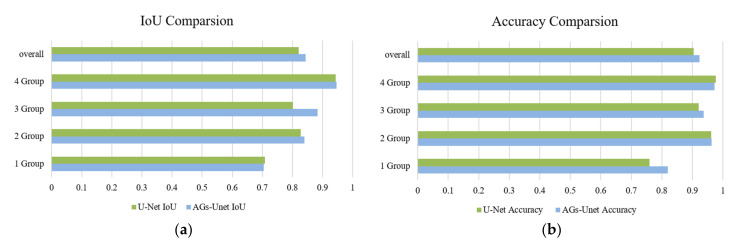
Comparison of *IoU* and *Accuracy* between U-Net and AGs-Unet models. The left figure (**a**) shows the comparison results of IoU between U-Net and AGs-Unet models while the right figure (**b**) shows the comparison results of Accuracy between U-Net and AGs-Unet models.

**Figure 8 sensors-22-02932-f008:**
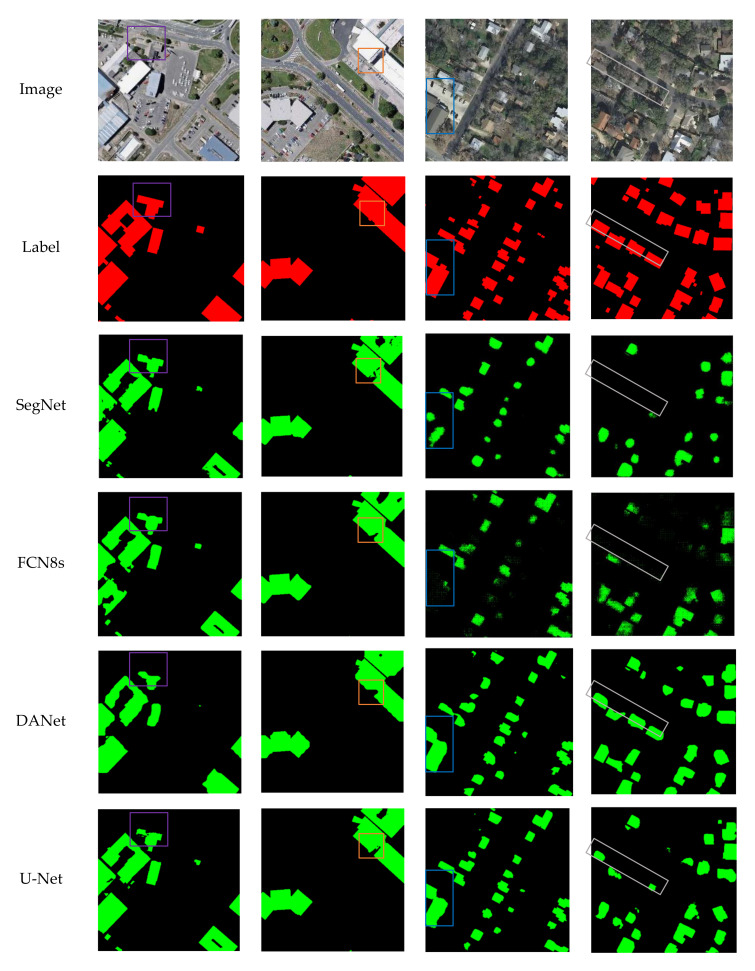
Comparison of extraction results of each model building in test dataset. The first two rows are aerial images and ground truth, respectively. Rows 3–8 are building extraction results of SegNet, FCN8s, DANet, U-Net, PISANet, ARC-Net, and our proposed AGs-Unet, respectively. The green and black pixels of the maps represent the predictions of true positive and true negative, respectively.

**Figure 9 sensors-22-02932-f009:**
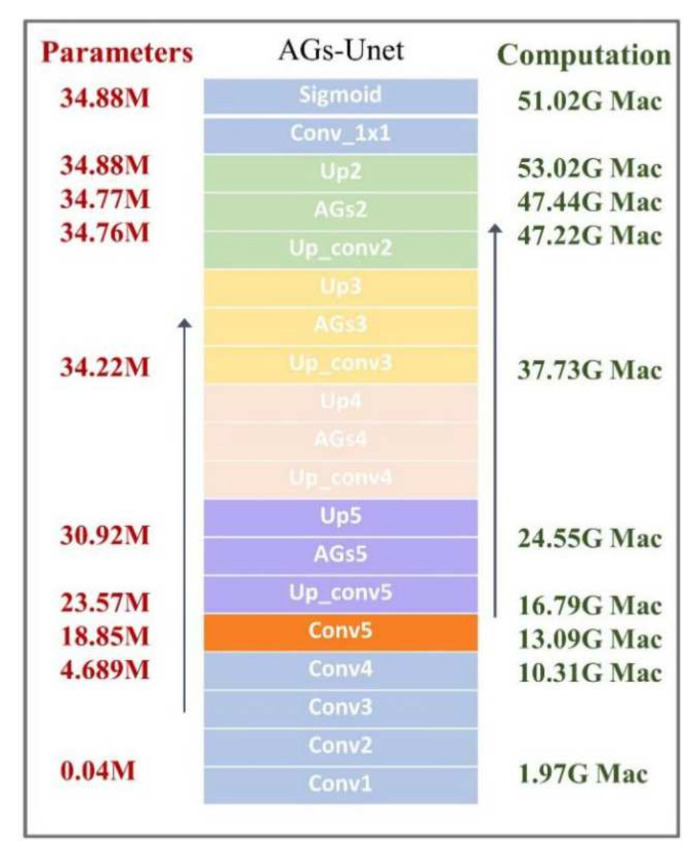
Statistics of structural parameters and calculations for AGs-Unet.

**Figure 10 sensors-22-02932-f010:**
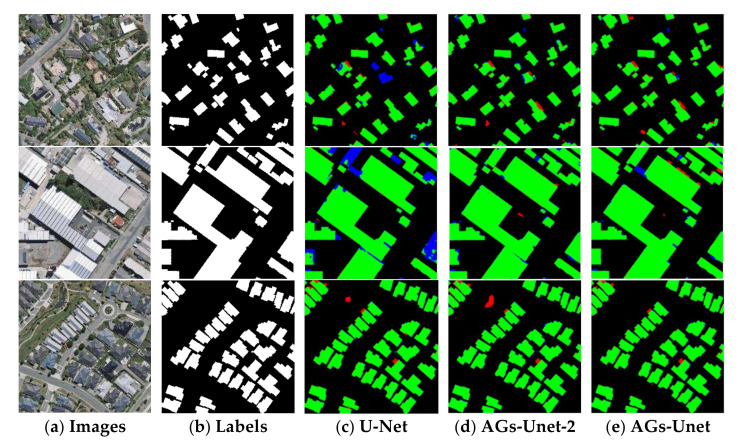
Comparison of aBlation experimental results. The first two columns are aerial images and ground truth, respectively. Columns 3–5 are building extraction results of U-Net, our proposedAGs-Unet-2 with 2 AG in U-Net, and AGs-Unet, respectively. The green, red, blue, and black pixels of the maps represent the predictions of true positive, false positive, false negative, and true negative, respectively.

**Table 1 sensors-22-02932-t001:** Detailed blocks of the proposed AGs-Unet outlined in Figure 1.

Block	Type	Filter	Channel Size	Output Size
				256 × 256 × 3
1	Conv1	(3, 3)	3 → 64	256 × 256 × 64
	Maxpool1	(2, 2)	64 → 64	128 × 128 × 64
2	Conv2	(3, 3)	64 → 128	128 × 128 × 128
	Maxpool2	(2, 2)	128 → 128	64 × 64 × 128
3	Conv3	(3, 3)	128 → 256	64 × 64 × 256
	Maxpool3	(2, 2)	256 → 256	32 × 32 × 256
4	Conv4	(3, 3)	256 → 512	32 × 32 × 512
	Maxpool4	(2, 2)	512 → 512	16 × 16 × 512
5	Conv5	(3, 3)	512 → 1024	16 × 16 × 1024
6	Up_conv4	Conv-(2, 2)	1024 → 512	32 × 32 × 512
	AGs4		512 → 512	32× 32 × 512
	Up4	Up-(3, 3)	512 → 512	32 × 32 × 512
7	Up_conv3	Conv-(2, 2)	512 → 256	64 × 64 × 256
	AGs3		256 → 256	64 × 64 × 256
	Up3	Up-(3, 3)	256 → 256	64 × 64 × 256
8	Up_conv2	Conv-(2, 2)	256 → 128	128 × 128 × 128
	AGs2		512 → 512	128 × 128 × 128
	Up2	Up-(3, 3)	512 → 512	128 × 128 × 128
9	Up_conv1	Conv-(2, 2)	128 → 64	256 × 256 × 64
	AGs 1		64 → 64	256 × 256 × 64
	Up1	Up-(3, 3)	64 → 64	256 × 256 × 64
10	Conv_1 × 1	(1, 1)	64 → 1	256 × 256 × 1
	Sigmoid			256 × 256 × 1

Conv: convolution; Maxpool: the maximum pooling; Up: up-sampling, AGs: attention gates; Up_conv: up-sampling and convolution.

**Table 2 sensors-22-02932-t002:** Parameters of AG.

Conv Block	Layers	Filter	Channel Size
*W_g_*	Conv + BN	(1, 1)	*F_g_* → *F_int_*
*W_x_*	Conv + BN	(1, 1)	*F_l_* → *F_int_*
Activation	Relu	*W_g_* + *W_x_*	-
*P_s_*	Conv + BN	(1, 1)	*F_int_* → 1
*P* * _out_ *	BN	*F_int_* × P*_s_*	1 → *F_int_*

**Table 3 sensors-22-02932-t003:** U-Net IoU, U-Net Accuracy, AGs-Unet IoU, and AGs-Unet Accuracy Statistics.

Groups	1	2	3	4
U-Net Accuracy	0.759	0.961	0.921	0.977
AGs-Unet Accuracy ^1^	0.819	0.963	0.937	0.973
U-Net IoU	0.709	0.827	0.802	0.944
AGs-Unet IoU	0.715	0.840	0.883	0.946

^1^ The first and second lines represent accuracy, and the third and fourth lines represent IoU. The maximum value is displayed by blackening.

**Table 4 sensors-22-02932-t004:** Quantitative evaluation results of the three indexes of each model in the WHU dataset and the INRIA dataset. The best scores are highlighted in bold.

Dataset	WHU Dataset			INRIA Dataset		
Model	*OA*	*Precision*	*IoU*	*OA*	*Precision*	*IoU*
SegNet	0.944	0.856	0.775	0.888	0.791	0.413
FCN8s	0.948	0.870	0.776	0.838	0.703	0.484
DANet	0.952	0.922	0.790	**0.929**	0.839	0.676
U-Net	0.967	0.931	0.843	0.916	0.862	0.671
PISANet	0.962	0.935	0.853	0.906	0.885	0.651
ARC-Net	0.952	0.855	0.793	0.921	0.835	0.679
AGs-Unet	**0.969**	**0.937**	**0.855**	0.919	**0.907**	**0.682**

**Table 5 sensors-22-02932-t005:** Parameters, computation, and training time of each model in WHU dataset and INRIA dataset. The highest scores are highlighted in bold.

Model	Parameters (M)	Computation (G Mac)	WHU Dataset Training Time (s)/Epoch	INRIA Dataset Training Time (s)/Epoch
SegNet	16.31	23.77	222	69
FCN8s	**134.27**	**62.81**	**393**	**74**
DANet	49.48	10.93	138	70
U-Net	13.4	23.77	212	66
PISANet	11.03	23.89	294	70
ARC-Net	16.19	16.6	193	69
AGs-Unet	34.88	51.02	316	72

**Table 6 sensors-22-02932-t006:** Accuracy statistics of ablation experiment in WHU dataset. The best scores are highlighted in bold.

Model	*OA*	*Precision*	*IoU*
U-Net	0.9713	**0.9689**	0.8358
AGs-Unet-2	0.9778	0.9251	0.8402
AGs-Unet	**0.9781**	0.9051	**0.8438**

## Data Availability

The link to download the WHU dataset can be found in the online version, at http://gpcv.whu.edu.cn/data/building_dataset.html (accessed on 18 March 2021).

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
