# Peer review of "AGs-Unet: Building Extraction Model for High Resolution Remote Sensing Images Based on Attention Gates U Network"

_sensors, 2022, doi:10.3390/s22082932_

Round 1
Reviewer 1 Report
This paper proposes the AGs-Unet, which can learn different forms of building structures in high-resolution remote sensing images and realize efficient extraction of building contour. Experimental results show that the proposed AGs-Unet model can improve the quality of building extraction from high-resolution remote sensing images effectively in terms of prediction performance and result accuracy. However, there are still some questions with suggestions to the authors:
- In the Introduction, the logic of this part of the end-to-end extraction is not clear, the summary of existing methods is not sufficient, and it is recommended to modify.
- What is the scientific question addressed by this article?
- What is the contribution and innovation of this article?
- What is the role of AGs Module? What is the visualization result of the attention map in the AGs Module?
- What is the difference between Unet、AGs-Unet-2 and AGs-Unet?
- In the Ablation Experiment, the comprehensive performance of AGs-Unet-2 is better than AGs-Unet, why?
- Using only one dataset is not enough to demonstrate the effectiveness and generalization of the model. It is necessary to add some experiments on other datasets.
- It is necessary to compare with more state-of-the-art methods to prove the effectiveness of the proposed method, such as latest building extraction models.
Reviewer 2 Report
Review form for the paper:
AGs-Unet: Building Extraction Model for High Resolution Remote Sensing Images Based on Attention Gates U Network
The introduction provides well-documented reference regarding the current research on image segmentation by various techniques like CNN or FCN
The methodology is accurate starting with AGs-Unet model with detail on each step and continuing with the attention mechanism and the usage of AG module for building extraction captures the attention in the rectified linear activation function.
The similitude with the human logic is not new but this approach with the activation of neurons related to the interest features.
Row 212: Small typos: im-age
Figure 3 We told about the label of the original image, my question outside the paper is: There is it also possible to learn the algorithm to identify not only the category buildings but also the type of buildings?
The usage of deep learning neural model like the Adaptive model of estimation based on hundreds of iterations are welcome
The result section with variate examples of satellite and aerial images from New Zealand is well structured and sustained by graphical images with classification; in all the situations compared with IoU and U-Net the results for AGs-Unet are superior.
The discussion section brings a comparison of different classification models with the emphasis of AGs-Unet advantage but unfortunate with a huge lack of reference from other research. I recommended of improvement of this chapter with the citation from the newest and advanced results from other experiments for each model introduced in comparison.
In the Conclusion chapter, I would like also to see the limitation of this algorithm, where it fails, and how can be improved.
My big question is why you submitted this nice paper to the section Radar Sensors? there is no use of radar data.
Reviewer 3 Report
The manuscript entitled “AGs-Unet: Building Extraction Model for High Resolution Remote Sensing Images Based on Attention Gates U Network” presents a model for building extraction from HR RS images.
The originality aspects of the manuscript are not mentioned.
Achievements should be highlighted in the abstract.
The study area and type of high resolution images are not specified.
Training and check data ratio is not mentioned.
Is the GSD effective on the results of the proposed method?
The manuscript is written well, however, some typos such as in line 223 "im-portance" should be corrected.
In order to verify the proposed method on different datasets by the readers, the implemented code should be accessed.
